# Experimental Study of Shear Performance of High-Strength Concrete Deep Beams with Longitudinal Reinforcement with Anchor Plate

**DOI:** 10.3390/ma16176023

**Published:** 2023-09-01

**Authors:** Shu-Shan Li, Tian-Cheng Jin, Li-Ang Zheng, Guang-Yao Zhang, Hong-Mei Li, Ai-Jiu Chen, Wei Xie

**Affiliations:** 1School of Civil Engineering and Communication, North China University of Water Resources and Electric Power, Zhengzhou 450046, China; lishushan@ncwu.edu.cn (S.-S.L.); 201305628@stu.ncwu.edu.cn (L.-A.Z.); zgyncwu@163.com (G.-Y.Z.); lihongmei@ncwu.edu.cn (H.-M.L.); caj@ncwu.edu.cn (A.-J.C.); xwei@ncwu.edu.cn (W.X.); 2Engineering Technology Research Center for Structural Vibration Control and Health Monitoring of Henan Province, Zhengzhou 450046, China

**Keywords:** deep beam, longitudinal reinforcement with an anchor plate, high-strength concrete, shear span ratio, reinforcement ratio, shear bearing capacity

## Abstract

As a transfer member at the discontinuous place of vertical load, the deep beam has a complex stress mechanism and many influencing factors, such as compressive strength of concrete, shear span ratio, and reinforcement ratio. At the same time, the stress analysis principle of traditional shallow beams is no longer applicable to the design and calculation of deep-beam structure. The main purpose of this paper was to use the strut-and-tie model to analyze its stress mechanism, and to verify the applicability of the model. Nine high-strength concrete deep-beam specimens with longitudinal reinforcement with an anchor plate of the same size were tested by two-point concentrated loading method. The effects of shear span ratio (0.3, 0.6, and 0.9), longitudinal reinforcement ratio (0.67%, 1.05%, and 1.25%), horizontal reinforcement ratio (0.33%, 0.45%, and 0.50%), and stirrup reinforcement ratio (0.25%, 0.33%, and 0.50%) on the failure mode, deflection curve, characteristic load, crack width, steel bar, and concrete strain of the specimens were analyzed. The results showed that the failure mode of deep-beam specimens was diagonal compression failure. The normal section cracking load was about 15 to 20% of the ultimate load, and the inclined section cracking load was about 30~40% of the ultimate load. The shear span ratio increased from 0.3 to 0.9, and the bearing capacity decreased by 32.9%. When the longitudinal reinforcement ratio increased from 0.67% to 1.25%, the ultimate load increased by 42.6%. The shear span ratio and longitudinal reinforcement ratio have a significant effect on the bearing capacity of the high-strength concrete deep beams with longitudinal reinforcement with an anchor plate. The shear capacity of nine high-strength concrete deep-beam specimens with longitudinal reinforcement with an anchor plate was calculated by national standards, and the results were compared with the calculation results of the Tan–Tang model, the Tan–Cheng model, SSTM, and SSSTM. The analysis showed that the softened strut-and-tie model takes into account the softening effect of compressive concrete, and is a more accurate mechanical model, which can be applied to predict the shear capacity of high-strength concrete deep-beam members with longitudinal reinforcement with an anchor plate.

## 1. Introduction

Deep beams are generally used as transfer components at discontinuous vertical loads in buildings and bridge structures. The stress mechanism of deep beams is complex and there are many factors affecting the bearing capacity, such as compressive strength of concrete, shear span ratio, longitudinal reinforcement ratio, and web reinforcement ratio. Many scholars [1,2,3,4,5,6,7] have conducted theoretical analysis and experimental research on the mechanical performance of deep-beam structures, studying the influence of these factors on the bearing capacity of deep beams. According to the Saint Venant’s principle, there is a significant difference in the mechanical performance of the stress discontinuity zone (referred to as the D zone) in deep beams compared to shallow beams [8]. The plane-section assumption and the traditional shallow-beam stress analysis principle are no longer applicable to the design and calculation of the deep-beam structure, and the design and calculation method of the deep beam has not been unified [9]. Therefore, specific calculation methods need to be used for deep-beam components [10].

Previous studies have shown that the shear span ratio is one of the important influencing factors on deep-beam components [11]. Tan et al. [12] proposed that the shear span ratio affects the failure mode of deep flexural members by reflecting the distribution of shear force and bending moment. Oh and Shin [13] proposed that the ultimate shear capacity of the component is controlled by the shear span ratio regardless of the strength of the concrete. Chen et al. [14] found that the shear capacity of steel-reinforced concrete deep beams decreased with the increase in shear span ratio. Jin [15] found that the shear span ratio had a great influence on the shear strength of the beam. The smaller the shear span ratio, the higher the shear strength of the deep beam. Tan et al. [16] analyzed the effect of web reinforcement on the shear behavior of deep beams. It is considered that when the shear span ratio is greater than 1, the improvement of the shear bearing capacity of stirrup reinforcement is more significant than that of horizontal reinforcement. When the horizontal and stirrup reinforcement are configured at the same time, the influence on the shear bearing capacity of deep flexural members is the most significant. At the same time, web reinforcement can effectively control the development of oblique cracks. Singh, et al. [17] found that by increasing the horizontal reinforcement ratio, the strut coefficient of ordinary strength and high-strength concrete increased by 31% and 25%, respectively. However, for higher reinforcement ratios, the increase in the strut coefficient was relatively small. Mihaylov et al. [18] found that stirrups increased the shear capacity by increasing the height of the top compression zone. Chinese code [19] also proposes that when the shear span ratio is large, the influence of web reinforcement is more obvious. With the decrease of span–depth ratio, the shear force of members changes from being mainly borne by stirrup reinforcement to only being borne by horizontal reinforcement. Regarding the influence of longitudinal reinforcement on deep-beam components, Tan et al. [12] found that when the shear span ratio was less than 1.5, the longitudinal reinforcement ratio had a significant impact on the shear bearing capacity of deep bending components. Liu et al. [20] found, through experiments, that as the reinforcement ratio increased, the shear bearing capacity of the component increased linearly, and this effect was obvious when the shear span ratio and span height ratio were small. Shi et al. [21] found that the anchor plate can give full play to the yield performance of the steel bar, so that the beam with the anchor plate has a larger influence area on the compression bar and improves the shear capacity of the component. Miao [22] and Sim [23] found the failure mode and bearing-capacity calculation method of steel bars with anchor plates. This fully demonstrated that the anchor plate can be used for anchoring deep beams and is beneficial for improving the shear bearing capacity of deep beams.

In the face of different influencing factors, the calculation methods of shear capacity of deep-beam members can be classified into two categories: one is the empirical-formula method based on experimental research results and statistical theory; one is a mechanical model based on the force mechanism of deep beams, such as the strut-and-tie model. The Chinese Code for Design of Concrete Structures GB50010-2010 [19] is based on the research results of the former, and the design method of shear capacity of deep beams is given. The empirical-formula method was mostly established by scholars according to the experimental research parameters, which had good agreement. At present, such methods seldom consider the influence of the longitudinal reinforcement ratio, section height, or plate (or column) size [24]. The other calculation method of shear capacity of deep beams is based on the strut-and-tie model (STM) by combining the mechanical model with the influencing parameters. This kind of model is based on the truss model, which was first proposed by Mitchell [25] and Mau [26]. Since then, its rationality has been confirmed, and its design ideas have also been applied to different specifications and model designs [27,28,29]. Since then, this study has been continuously revised and improved. Tan et al. [30,31] simplified the bearing capacity model of deep beams into the arch-and-tie model, and considered the influence of size effect, and proposed the Tan–Tang model and the Tan–Cheng model. Hwang et al. [32] proposed the softened strut-and-tie model (SSTM) considering the softening effect of concrete on the basis of the strut-and-tie model, and then simplified the model to propose the simplified softened strut-and-tie model (SSSTM) [33]. Wu et al. [34] also put forward the crack band theory on this basis. Kondalraj [35] also modified the strut coefficient of ACI318-19 based on the STM model and proposed it as a function of concrete strength.

At present, domestic and foreign scholars have conducted a large amount of research on deep bending components, but their influencing factors and data dispersion are relatively large. There is little research on the system of high-strength concrete deep-beam components with longitudinal reinforcement with anchor plates. It is of practical significance to study the mechanical performance mechanism of high-strength concrete deep beams with longitudinal reinforcement with anchor plates, and propose reasonable design methods and bearing-capacity calculation formulae. Therefore, this article describes the design and manufacture of nine high-strength concrete deep-beam components with anchor plate longitudinal reinforcement; analyzes the influence of shear span ratio, longitudinal reinforcement ratio, and web reinforcement ratio on the failure mechanism of the components; analyzes their failure process and mode; and discusses the applicability of various calculation methods, in order to provide theoretical reference for the practical engineering application of such components.

## 2. Test Overview

### 2.1. Specimen Design

In order to study the influence of shear span ratio, longitudinal reinforcement ratio, horizontal reinforcement ratio, and stirrup reinforcement ratio on the bearing capacity of high-strength concrete deep beams with longitudinal reinforcement with an anchor plate, a total of nine specimens were designed and manufactured. The size was 1660 mm × 600 mm × 200 mm, and the specimens were divided into Groups a–d (Group a took the shear span ratio as the variable, and included DBH-1, DBH-2, and DBH-3. Group b took the longitudinal reinforcement ratio as the variable, and included DBH-2, DBH-4, and DBH-5. Group c took the horizontal reinforcement ratio as the variable, and included DBH-2, DBH-6, and DBH-7. Group d took the stirrup reinforcement ratio as the variable, and included DBH-2, DBH-8, and DBH-9), according to the different influencing factors. Specimen DBH-2 was the control specimen. The detailed reinforcement information and design parameters of the specimens are shown in Table 1 and Figure 1. In order to prevent dense anchorage of longitudinal reinforcement and give full play to the tensile efficiency of longitudinal reinforcement, reduce the cost of steel used for anchorage, facilitate construction, and overcome the difficulty of concrete pouring, the end of longitudinal reinforcement was anchored by the anchor plate. The net bearing area of the anchor plate was four times larger than that of longitudinal reinforcement. The real image of the reinforcement with anchor plates is shown in Figure 2.

### 2.2. Experiment Material

In this experiment, nine specimens were poured with C60 concrete. At the same time, six groups of the same batch of pouring test blocks were reserved, and the specimens were cured with the same curing conditions until the beginning of the test to ensure the accuracy of the material test. According to the relevant provisions in GB/T50152-2012 [36] and GB/T50081-2012 [37], the material test of the test block was carried out, and the material performance test results are shown in Table 2.

In this test, the bottom longitudinal reinforcement was an HRB600 steel bar with diameter of 16 mm, 20 mm, and 22 mm, and the web reinforcement was an HRB400E steel bar with diameter of 8 mm. The material test of steel bar is determined according to the relevant provisions of GB/T228.1-2010 [38], and the material performance test results are shown in Table 3.

### 2.3. Loading System and Data Collection

Firstly, the bearing capacity of the specimen was estimated. The single jack device was used to carry out symmetrical concentrated loading through the distribution beam. The size of the plate at the loading point and the support was 80 mm × 200 mm, and the gap between the plate and the deep beam was filled with fine sand to ensure the average force on both sides. The loading device of the test is shown in Figure 3. Before cracking, 50 kN was used as incremental load for each level of the specimen, and 100 kN was used as incremental load after cracking. The loading rate was 2 kN/s, and the load was held for 2 min after each level was loaded to observe and record the test data. When the specimen was damaged or the bearing capacity was reduced to 70% of the peak load, the loading was completed. The loading-system curve is shown in Figure 4.

The main data collected during the loading process are as follows: (1) deflection: three displacement meters were arranged at the middle position of the bearing and the bottom of the beam to measure the vertical displacement of the specimen; (2) characteristic load: including normal section cracking load, inclined-section cracking load and ultimate load; (3) concrete strain: oblique and vertical strain gauges were arranged evenly on both sides of the specimen and the vertical part of the beam span to measure the concrete strain; (4) longitudinal reinforcement strain: strain gauges were arranged at equal intervals on the bottom longitudinal reinforcement to measure the strain of the bottom longitudinal reinforcement and clarify the effect of the longitudinal reinforcement; (5) strain of web reinforcement: strain gauges were arranged at equal intervals on the web reinforcement in the shear span area to measure the strain of web reinforcement and clarify the shear resistance of web reinforcement; and (6) crack width development: the occurrence of cracks and the width of existing cracks were recorded after each stage of loading. The arrangement of displacement meters and strain gauges on the specimen is shown in Figure 5.

## 3. Experiment Results and Analyses

### 3.1. Experiment Failure Process

The process from loading to failure is basically the same, and can be divided into the elastic stage, normal-section cracking stage, inclined-section cracking stage and ultimate failure stage. From the beginning of formal loading to the first normal-section crack, the specimen was in elastic working condition. There was no obvious deformation or crack on the surface of the specimen, and the strain of the steel bar was small. The stress state of the specimen conformed to the elastic state, and the specimen was in the elastic stage. As the load continued to increase, the first bending crack appeared near the mid-span, with a width of 0.02 mm–0.07 mm, and the specimen entered the normal-section cracking stage. After that, the cracks continued to develop, and multiple cracks appeared near the first crack. The crack density increased, the height gradually extended to 1/6–1/3 of the beam height, the crack extension angle gradually pointed to the mid-span position at the top of the beam, the crack width increased slowly, and the bending moment in the shear span area increased. Group a specimens showed that the crack development was due to the bending effect at this time, and the bending effect became more and more obvious with the increase in the shear span ratio. The crack generation and development were caused by the tension at the bottom of the beam. When the load reached 30% to 40% of the ultimate load, the first diagonal crack appeared at the support, and the crack developed rapidly from the bottom to the top along the connection between the support and the loading plate on the same side. Then, diagonal cracks appeared at the same position on the other side of the specimen, and the specimen entered the diagonal-section cracking stage. The height of the mid-span bending crack eventually reached half of the beam height as the load increased, and individual specimens reached 2/3 of the beam height. Under the influence of bending and shearing in the shear span area, the cracks continued to extend, and multiple parallel cracks appeared near the cracks. The crack density in the compression bar area increased and the cracks penetrated each other, and the crack width increased. After that, the crack development rate decreased, and the stress state of the specimen tended to be stable, forming an obvious force transmission mechanism of the STM. After that, the concrete had a splitting sound as the load increased, and the specimen entered the ultimate failure stage. Finally, with a loud noise, the specimen was destroyed, and a penetrating oblique crack was generated on one side of the specimen. The concrete at the support at the bottom of the crack was partially broken, and the crack width was also increased to 0.7 mm–1.6 mm. The specimen had obvious brittle failure.

### 3.2. Experiment-Failure Mode

Taking the specimen DBH-2 as an example, the specimen showed a typical failure mode as a control specimen, as shown in Figure 6. From the Figure 6, it can be seen that the failure surface was located in the concrete strut at the connection between the left support and the backing plate, and was a diagonal compression failure in the form of concrete-strut crushing.

Figure 7 shows the final crack failure mode of each group of specimens. The fracture morphology of each group of specimens was similar. All were crushed by the concrete strut and had the obvious force transmission mechanism of the STM. It can be seen from the crack morphology that the degree of crack development showed a significant positive correlation with the shear span ratio. As the shear span ratio increased, the crack density and crack extension height at the bottom of the beam span increased. This was because the failure of the deep beam depended on the internal force of the component, that is, the shear force and the bending moment. The change in the shear span ratio changes the distribution of the shear force and the bending moment, so that the failure mode changed from shear compression failure to bending failure. In this experiment, the reinforcement ratio of the bottom longitudinal reinforcement had little effect on the crack development. The main reason was that the bottom longitudinal reinforcement acted as a tie rod in the STM. In this group of specimens, the bottom longitudinal reinforcement did not reach the yield state, and the tie rod did not give full play to the effect, so the failure pattern was similar. The crushed strut bar in the specimen showed obvious concrete spalling at the loading point or the support. However, as the reinforcement ratio of the web reinforcement increased, the depth of the spalling surface decreased, and the bending cracks at the bottom of the beam span developed more fully. This was because the web reinforcement could be used to increase the ductility of the beam by enhancing the aggregate bite cooperation of the concrete abdomen, indicating that the web reinforcement was beneficial to the development of cracks. The specimens in each group made a splitting sound when they were close to failure, and many parallel cracks appeared in the strut bar. The failure was accompanied by loud noise, showing obvious brittle failure.

### 3.3. Experiment Damage Results

The failure results of each group in the experiment are shown in Table 4. $V_{cr}^{N}$ is defined as the load corresponding to the first crack in the mid-span, $V_{cr}^{D}$ is defined as the load corresponding to the first crack in the inclined section, $V_{u}$ is defined as the ultimate load of the specimen, and δ is the maximum deflection in the mid-span. “ Load ” refers to the vertical load on one side of the component.

The characteristic load curve of Figure 8 shows that the cracking load of the normal section was about 15–20% of the ultimate load except Group a of specimens, and had no obvious correlation with the reinforcement ratio of the steel bar. The reason for this was that the load of the specimen was mainly borne by the concrete part and the steel bar did not bear a load, or the load was small. The $V_{cr}^{N}$/$V_{u}$ of Group a decreased with the increase in shear span ratio. The reason for this was that the increase in shear span ratio meant that the specimen as subjected to an obvious bending effect, which was more likely to produce bending cracks.

Compared with the cracking load of the inclined section in each group, the cracking load of the inclined section in the same group changed little, but the increase in the horizontal reinforcement ratio and the longitudinal reinforcement ratio decreased the $V_{cr}^{D}$/$V_{u}$, indicating that the specimen still had sufficient safety margin after cracking. The reason for this was to increase the reinforcement ratio in the horizontal direction, increase the overall bond stress between the steel bar and the concrete, effectively restrain the lateral strain of the concrete, and subject the horizontal steel bar to force after cracking. The horizontal reinforcement could be regarded as a horizontal tie rod. The increase in the reinforcement ratio increased the effective cross-sectional area of the tie rod, thereby increasing the ultimate load of the specimen and reducing $V_{cr}^{D}$/$V_{u}$.

Compared with the ultimate load in each group, the ultimate load decreased with the increase in shear span ratio. Compared with λ = 0.3, the ultimate load decrease by 18.5% when λ = 0.6, and decreased by 32.9% when λ = 0.9. The reason for this was that the inclination angle of the strut bar decreased with the increase in the shear span ratio, which mad the shear force longer along the force-transmission path of the strut bar, reducing the force transmission mechanism of the STM, and reducing the ultimate bearing capacity. The ultimate load increased with the increase in longitudinal reinforcement ratio. When the longitudinal reinforcement ratio increased from 0.67% to 1.05% and 1.25%, the ultimate load increased by 18.2% and 42.6%, respectively. This is consistent with the conclusion in [39]. The reason was that the longitudinal reinforcement could improve the bearing capacity of the deep beam through the dowel action. The longitudinal reinforcement could inhibit the development of the inclined crack, and could also improve the performance of the shear force transfer between the inclined-crack interfaces. At the same time, the longitudinal reinforcement was used as the tie bar in the STM. Increasing the reinforcement ratio of the longitudinal reinforcement increased the cross-sectional area of the tie bar, thereby increasing the strength of the tension bar and significantly increasing the ultimate load. The ultimate load increased with the increase in reinforcement ratio, but its effect was not obvious. The reason was that the reinforcement skeleton, composed of web reinforcement, improved the ductility of the specimen by effectively restraining the strain of concrete and the expansion of cracks [40], thus improving the bearing capacity, but this improvement was limited.

### 3.4. Mid-Span Deflection Curve

The mid-span deflection curve of each group in the test is shown in Figure 9. From the initial stage of loading to the first crack, the curves showed an upward trend and the slopes were basically the same, indicating that the initial stiffness of the specimen was basically the same and was in the elastic working stage. When the first crack appeared, the slope of the curve was significantly reduced, and the stiffness was degraded. The reason for this was that the appearance of the crack releases the internal energy of the specimen, and the specimen is in the elastic–plastic stage. After that, the load continued to increase, the slope of the curve was basically unchanged, and the specimen was in a stable state. When the load increased to the appearance of through cracks, the slope of the curve decreased again, and the stiffness also decreased. When the load reached the ultimate load, the specimen was destroyed.

From Figure 9a, it can be concluded that the deflection increased with the increase in shear span ratio. When the shear span ratio was 0.3, 0.6, and 0.9, the maximum mid-span deflection was 2.19 mm, 3.56 mm, and 4.39 mm, respectively. This is because as the shear span ratio increased, the failure mode of the specimen gradually changed to bending failure, and the number and width of cracks at the bottom of the mid-span increased, which eventually led to faster release of internal energy and a significant reduction in stiffness, resulting in an increase in mid-span deflection. From Figure 9b, it can be seen that the deflection dd not show significant difference when the load was below 500kN. after that, the deflection decrease slightly with the increase in the longitudinal reinforcement ratio at the bottom. When the longitudinal reinforcement ratio was 0.67%, 1.05%, and 1.25%, the maximum mid-span deflection was 3.16 mm, 3.56 mm, and 3.55 mm, respectively. This was due to the development of the height and width of the crack. The bottom longitudinal reinforcement began to participate in the force to form a tie rod. The increase in the reinforcement ratio of the bottom longitudinal reinforcement increased the stiffness of the specimen and increased the bending resistance of the specimen so that the mid-span deflection was reduced. However, the bottom longitudinal reinforcement did not yield when the specimen reached the ultimate load, and did not give full play to its role, so the maximum mid-span deflection was similar. Comparing Figure 9c,d, when the horizontal reinforcement ratio was 0.33%, 0.45%, and 0.50%, the maximum mid-span deflection was 3.56 mm, 4.01 mm, and 4.35 mm, respectively. When the stirrup reinforcement ratio was 0.25%, 0.33%, and 0.50%, the maximum mid-span deflection was 4.31 mm, 3.56 mm, and 3.88 mm, respectively. Although the change of web reinforcement ratio can limit the development of inclined cracks, it had no obvious effect on the bending resistance of the specimen. Therefore, the change of web reinforcement ratio had no obvious effect on the maximum mid-span deflection of the specimen.

### 3.5. Crack-Width Curve of Oblique Section

The crack-width curve of each group of oblique sections in the experiment is shown in Figure 10. It can be seen from Figure 10a that with the increase in shear span ratio, the earlier the crack appeared, the larger the crack width, and the lower the slope of the curve. When the shear span ratio was 0.3, 0.6, and 0.9, the maximum width of the inclined-section crack was 0.72 mm, 0.98 mm, and 1.60 mm, respectively. The reason for this was that the decrease of the inclination angle of the compression bar changed the stress distributed on the compression bar and the tension bar. The compressive stress at the support decreased, and the tensile stress of the tie bar increased. The specimen was subjected to bending, and it was difficult to resist the emergence and development of cracks. It can be seen from Figure 10b that the longitudinal reinforcement had little effect on the crack width when the load was small. When the load was large, the crack width decreased with the increase in the longitudinal reinforcement ratio. When the longitudinal reinforcement ratio was 0.67%, 1.05%, and 1.25%, the maximum crack widths of the inclined sections were 0.76 mm, 0.98 mm, and 0.60 mm, respectively. After the formation of inclined cracks, the deep-beam members gradually formed an STM system with longitudinal reinforcement as the tie bar and concrete between the loading plate and the supporting plate as the compression bar. In particular, the longitudinal reinforcement at the support limited the development of inclined cracks by pinning action. The increase in longitudinal reinforcement led to the decrease of diagonal crack width, which enhanced the interlocking effect of aggregate, thus improving the shear capacity of deep beams. It can be seen from Figure 10c,d that when the horizontal reinforcement ratio was 0.33%, 0.45%, and 0.50%, the maximum width of inclined-section cracks is 0.98 mm, 0.97 mm and 0.97 mm, respectively. When the stirrup reinforcement ratio was 0.25%, 0.33%, and 0.50%, and the maximum width of the oblique section crack was 1.22 mm, 0.98 mm, and 0.98 mm, respectively. The reinforcement skeleton formed by the web reinforcement could effectively constrain the strain of the concrete, thereby reducing the crack width. The reason for the unobvious difference in the early stage of Figure 10c is that the extension height of the crack in the early stage was low, and the number of horizontal reinforcements passing through the crack was small. In the later stage of loading, the crack was fully developed, and more horizontal reinforcements participated in the tensile strength. Therefore, increasing the horizontal reinforcement ratio helped to reduce the crack width, which is consistent with Tan et al. [41] that if the horizontal and stirrup reinforcement are distributed at the same time, it has a better limiting effect on the development of oblique cracks in deep beams.

### 3.6. Steel-Strain Curve

The steel bars distributed in the deep beam were divided into horizontal reinforcement and longitudinal reinforcement. The longitudinal reinforcement affected the bearing capacity of the beam through the dowel action. The longitudinal stressed steel bar not only inhibited the development of the oblique crack, but it also improved the performance of the shear force transfer between the crack interface. At the same time, the longitudinal reinforcement acted as a tie rod in the STM, which is one of the important components in the model. In addition to directly bearing part of the shear force, the horizontal reinforcement indirectly limited the development of cracks and enhanced the aggregate bite force of the abdominal concrete. The horizontal reinforcement passing through the diagonal compression reinforcement improved the strength of the concrete diagonal compression bar and also affected the bearing capacity of the deep beam.

#### 3.6.1. Average-Strain Curve of Longitudinal Reinforcement

The longitudinal reinforcement strain curve of each group in the experiment is shown in Figure 11. The strain of the longitudinal reinforcement was small before the cracking of the specimen, and the strain increased rapidly after the cracking, and there was a clear slope downward trend. The stress mechanism of the tie rod arch was formed inside the specimen. The longitudinal reinforcement acted as a tie rod, and the strain increase with the increase in the load, indicating the validity of the STM. It can be seen from Figure 11a that the increase in shear span ratio made the strain of longitudinal reinforcement at the bottom increase obviously. This was because the tie rod bore more tension, and the failure mode changed from bending shear failure to bending failure. Therefore, when the shear span ratio is small, the reinforcement ratio of longitudinal reinforcement can be appropriately reduced to achieve better economic benefits. It can be seen from Figure 11b that with the increase in longitudinal reinforcement ratio, the strain of longitudinal reinforcement decreased under the same load. The reason for this was that the increase in the longitudinal reinforcement ratio made the cross-sectional area of the tie rod large, resulting in the decrease of the longitudinal reinforcement strain. From Figure 11c,d, it can be seen that the increase in horizontal reinforcement ratio increased the strain of longitudinal reinforcement, and the stirrup reinforcement ratio had little effect on the strain of longitudinal reinforcement. The reason was that the horizontal reinforcement could effectively constrain the transverse strain of concrete, so that the strain of longitudinal reinforcement and the strain of bottom concrete tended to be consistent, and the stirrup had no effect on the constraint transverse strain. Therefore, increasing the horizontal reinforcement ratio can give full play to the role of longitudinal reinforcement.

#### 3.6.2. Average-Strain Curve of Horizontal Reinforcement

The strain curves of horizontal bars in each group are shown in Figure 12. The shear span ratio had a significant effect on the horizontal reinforcement. With the increase in the shear span ratio, the earlier the horizontal reinforcement reached the yield, and the increase in the shear span ratio increased the transverse tensile strain of the concrete, resulting in the greater tensile stress on the horizontal reinforcement. Figure 12b,c has similar rules but different reasons. In Group b, the tensile stress of the specimen was shared by the horizontal reinforcement and the longitudinal reinforcement. The increase in the longitudinal reinforcement ratio made the longitudinal reinforcement bear more tensile stress, so the horizontal reinforcement strain decreased with the increase in the longitudinal reinforcement ratio. In Group c, the increase in the horizontal reinforcement ratio made the cross-sectional area of the horizontal reinforcement increase, so the horizontal reinforcement strain decreased with the increase in the horizontal reinforcement ratio. The stirrup reinforcement ratio has almost no effect on the horizontal reinforcement strain.

### 3.7. Concrete-Strain Curve

The specimen DBH-2 is taken as an example, as shown in Figure 13. Before cracking, the concrete was in the elastic stage, the concrete strain was basically linear, and the concrete on both sides was in a state of compression, which conformed to the characteristics of the strut bar in the STM. After cracking, the strain of concrete increased rapidly, and some strain gauges were torn due to concrete spalling. There was no large crack in the concrete strut on the other side, and the strain continued to increase, which is consistent with the oblique compression failure of the strut crushing.

## 4. Results

### 4.1. Codes of Every Country

At present, the calculation methods of shear bearing capacity of deep flexural members in various countries are mainly divided into two categories: one is the semi-empirical and semi-theoretical formula obtained by regression of various influencing factors through big data, such as GB50010-2010 “Code for Design of Concrete Structures” [19], the other is the strut-and-tie model based on truss model (Figure 14), which is accepted by many countries because of its clear mechanical concept, such as ACI318-19 [27], European code [28], and CSAA23.3-19 [29]. The calculation models are shown in Table 5.

### 4.2. Tan–Tang Model and Tan–Cheng Model

Tan, K. H. et al. [30] proposed the Tan–Tang model based on the STM, and then proposed the Tan–Cheng Model based on the Tan–Tang model. The Tan–Cheng model considers the influence of the size effect law on the geometric size and boundary conditions of the concrete struts. The calculation model is detailed in Table 6.

### 4.3. SSTM and SSSTM

After cracking, the compressive strength of concrete has a significant decreasing trend. Cracked concrete produces a large tensile strain perpendicular to its compressive direction. These tensile strains have the opposite effect to the compressive strain (constraint effect), resulting in the strength of concrete being lower than that of concrete under the standard test. This phenomenon is called softening of concrete. The current national specifications for the application of the STM refer to the reduction coefficient $\beta_{s}$ for the reduction of the strength of the strut. Hwang et al. [32] proposed a SSTM for reinforced concrete deep beams based on previous studies. The SSTM not only satisfies the equilibrium condition and considers the softening of concrete, but also satisfies the constitutive equation and the deformation compatibility condition, which is more consistent with the actual situation. Since then, Hwang. [33] has simplified the iterative solution of the softening coefficient and proposed the SSSTM.

#### 4.3.1. SSTM

Macro model: Due to the bending moment balance equation, the relationship between horizontal shear force and vertical shear force can be determined, that is Equation (1)
(1)$$\frac{V_{bv}}{V_{bh}}=\frac{jd}{a}$$
where $V_{bv}$ is the vertical shear force of the deep beam; $V_{bh}$ is the horizontal shear force of the deep beam; *jd* is the distance between the center of mass of the compressive concrete of the deep beam and the center of mass of the tensile longitudinal reinforcement; and a is the horizontal distance from the load to the center of the bearing.

Force balance: According to the stress analysis, the horizontal shear force and vertical shear force in the deep beam are:(2)$$V_{bh}=-D\cos\theta +F_{h}+F_{v}\cot\theta$$
(3)$$V_{bv}=-D\sin\theta +F_{h}\tan\theta +F_{v}$$
where *D* is the pressure of the strut rod, $F_{h}$ is the tension of the vertical rod, and $F_{v}$ is the tension of the horizontal rod.

The maximum compressive stress $\sigma_{d,max}$ of concrete in the range of compressive bar of deep-beam member is:(4)$$\sigma_{d,max}=\frac{1}{A_{str}}\left[D-\frac{F_{h}}{\cos\theta }\left(1-\frac{\sin^{2}\theta }{2}\right)-\frac{F_{v}}{\sin\theta }\left(1-\frac{\cos^{2}\theta }{2}\right)\right]$$

Constitutive relation: The constitutive model of cracked concrete adopts Equations (5)–(7)
(5)$$\sigma_{d}=-\zeta f_{c}^{,}\left[2\left(\frac{-\varepsilon_{d}}{\zeta \varepsilon_{0}}\right)-{\left(\frac{-\varepsilon_{d}}{\zeta \varepsilon_{0}}\right)}^{2}\right],\frac{-\varepsilon_{d}}{\zeta \varepsilon_{0}}\leq 1$$
(6)$$\sigma_{d}=-\zeta f_{c}^{,}\left[1-{\left(\frac{-\varepsilon_{d}/\zeta \varepsilon_{0}-1}{2/\zeta -1}\right)}^{2}\right],\frac{-\varepsilon_{d}}{\zeta \varepsilon_{0}}>1$$
(7)$$\zeta =\frac{5.8}{\sqrt{f_{c}^{,}}}\frac{1}{\sqrt{1+400\varepsilon_{r}}}\leq \frac{0.9}{\sqrt{1+400\varepsilon_{r}}}$$

Deformation compatibility: The two-dimensional membrane element should satisfy the Mohr’s circular strain compatibility condition, Equation (8):(8)$$\varepsilon_{r}+\varepsilon_{d}=\varepsilon_{h}+\varepsilon_{v}$$
where $\varepsilon_{r}$ is the tensile strain of the spindle, $\varepsilon_{d}$ is the compressive strain of the spindle, $\varepsilon_{h}$ is the strain of the horizontal rod, and $\varepsilon_{v}$ is the strain of the vertical rod.

#### 4.3.2. SSSTM

The strut of reinforced concrete in the SSSTM is defined as $C_{d}$, and satisfies:(9)$${\;C}_{d}=\left(K_{h}+K_{v}-1\right)\zeta f_{c}^{,}A_{str}$$
(10)$$\zeta \approx 3.35/\sqrt{f_{c}^{,}}\leq 0.52$$
where $K_{h}$ and $K_{v}$ are horizontal and vertical rod indexes, respectively, and ζ is the softening coefficient of concrete.

The horizontal pull rod index is:(11)$$K_{h}=1+\left(\bar{K_{h}}-1\right)\times \frac{A_{th}F_{yh}}{\bar{F_{h}}}\leq \bar{K_{h}}$$
(12)$$\bar{K_{h}}\approx \frac{1}{1-0.2\left(\gamma_{h}+{\gamma_{h}}^{2}\right)}$$
(13)$$\bar{F_{h}}=\gamma_{h}\times \left(\bar{K_{h}}\zeta f_{c}^{,}A_{str}\right)\times \cos\theta$$

The vertical tie rod index is the same.

### 4.4. Results and Analysis

Nine deep-beam specimens were calculated by using the proposed method of national codes, the model proposed by Professor Tan, and the softened strut-and-tie model. The calculation results and comparative analysis are shown in Table 7 and Table 8 and Figure 15.

The mean value of the ratio of the experiment value to the calculation results of the method recommended by the GB50010-2010, ACI318-19, European code, and the CSAA23.3-19 was 1.529, 1.309, 1.313, and 1.473, respectively, and the variance was 0.043, 0.027, 0.026, and 0.022, respectively. It was shown that the ACI318-19 was close to the experiment results, which indicates the accuracy of the STM, and the Chinese code has a larger safety reserve.

The mean values of the ratio of the experimental values to the calculated results of Tan–Tang model and Tan–Cheng model were 1.359 and 1.318, respectively, and the variances were 0.007 and 0.006, respectively, indicating the accuracy of the model proposed by Professor Tan. The model takes into account the size effect and the data deviation was small.

The calculated results of the softened strut-and-tie model method were in good agreement with the experimental results. The average ratios of the experimental values to the calculated values of the softened strut-and-tie model and the simplified softened strut-and-tie model were 1.295 and 1.282, respectively, and the variances were 0.010 and 0.007, respectively. The softened strut-and-tie model takes into account the softening effect of compressive concrete, and is a more accurate mechanical model, which can accurately predict the shear capacity of high-strength concrete deep beam with anchor plate longitudinal reinforcement.

The calculation results of this paper were compared with the literature [4,5], and the calculation results tended to be consistent. The Chinese code uses the concrete item, the stirrup item, and the horizontal reinforcement item, but the experimental results showed that the increase in the longitudinal reinforcement ratio has an increasing effect on the bearing capacity of the deep beam, so the calculation results of the Chinese code are relatively conservative. Compared with the strut-and-tie model that only considers the strength of concrete in the strut, Professor Tan’s model comprehensively considers the strength of concrete in the strut and the effect of steel bars, so the data are more accurate and less discrete. The softened strut-and-tie model considers the conditions of force balance, constitutive relationship, and deformation coordination, and the calculation results are more accurate. However, compared with other scholars, it is found that this method does not consider the influence of scale effect for large-size specimens, and the results of large-size specimens may be unsafe.

## 5. Conclusions

In this paper, the failure mechanism and shear capacity calculation method of deep beams were analyzed by systematic experimental study on nine high-strength concrete deep-beam specimens with longitudinal reinforcement with an anchor plate. The Tan–Tang model, Tan–Cheng model, SSTM, and SSSTM based on the STM were used to calculate. The main conclusions are as follows:The failure process of the specimens was the same, and the failure mode was the diagonal compression failure of the concrete strut crushing, showing obvious brittle failure. The cracking load of the normal section was about 15~20% of the ultimate load, and the cracking load of the inclined section was about 30~40% of the ultimate load. The shear span ratio had a significant effect on the failure mode of deep beams.The shear span ratio is an important factor affecting the ultimate bearing capacity of deep beams. When the shear span ratio increased from 0.3 to 0.9, the bearing capacity decreased by 32.9%. The longitudinal reinforcement ratio had no obvious effect on the cracking load of the normal section of the specimen, but had a significant effect on the cracking load and ultimate load of the inclined section. When the longitudinal reinforcement ratio increased from 0.67% to 1.05% and 1.25%, the ultimate load increased by 18.2% and 42.6%, respectively. The increase in longitudinal reinforcement ratio can increase the bearing capacity of concrete.The reinforcement ratio of the web reinforcement will also have an impact on the bearing capacity of the deep beam. The steel skeleton formed by the combination of horizontal reinforcement and stirrups can effectively constrain the strain of the concrete, improve the ductility of the deep-beam member, and increase the effective cross-sectional area and strength of the concrete pressure bar, and the web reinforcement can also be used as a force-transmission path to improve the bearing capacity of the component. The increase in the reinforcement ratio of the web reinforcement can reduce the concrete strain, thereby reducing the width of the inclined crack. The web reinforcement is the same as the longitudinal reinforcement, which had little effect on the cracking load of the component, and the restraint effect on the concrete did not increase linearly;The shear bearing capacity of reinforced concrete deep-beam specimens was calculated by using the calculation method recommended by the specification and the Tan–Tang model, Tan–Cheng model, SSTM, and SSSTM based on the strut-and-tie model. The results showed that the ACI318-19 code was similar to the test results, and the Chinese code has a greater safety reserve. The model proposed by Professor Tan takes into account the size effect and the data deviation was small. The calculation results of the softened strut-and-tie model method were close to the experimental results. The softened strut-and-tie model considers the softening effect of the concrete strut bar, and has a clear mechanical calculation model, which can reasonably reflect the stress mechanism of the components.

## Figures and Tables

**Figure 1 materials-16-06023-f001:**
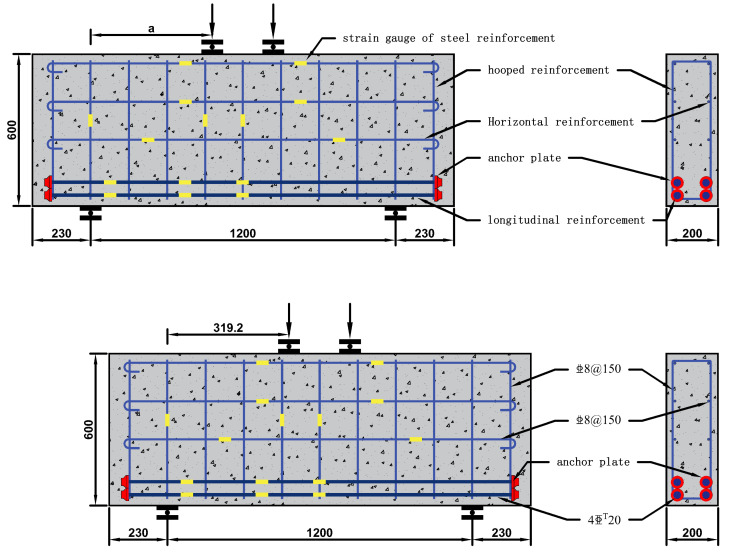
Specimen and DBH-2 detailed reinforcement information.

**Figure 2 materials-16-06023-f002:**
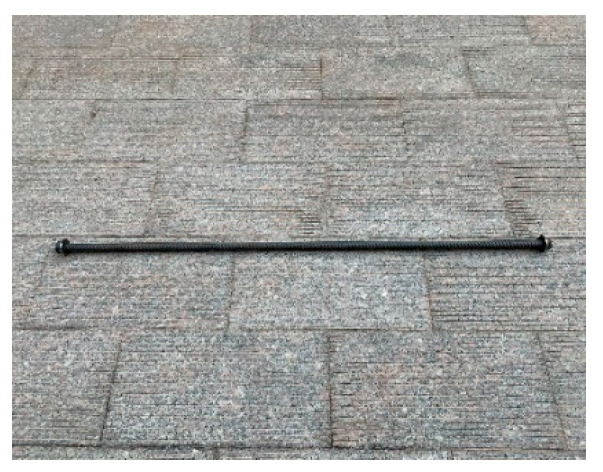
The real image of the reinforcement with anchor plates.

**Figure 3 materials-16-06023-f003:**
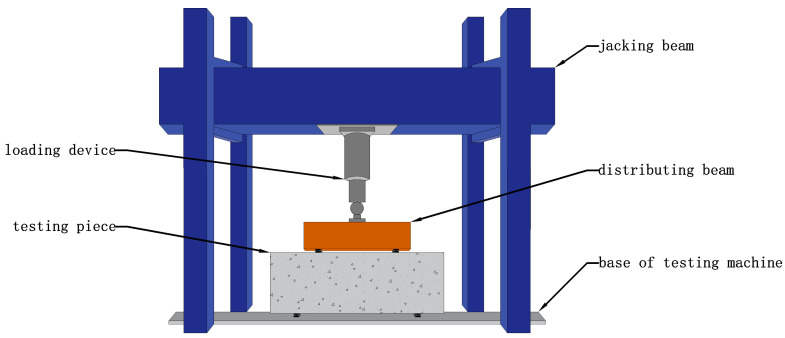
Experimental loading device.

**Figure 4 materials-16-06023-f004:**
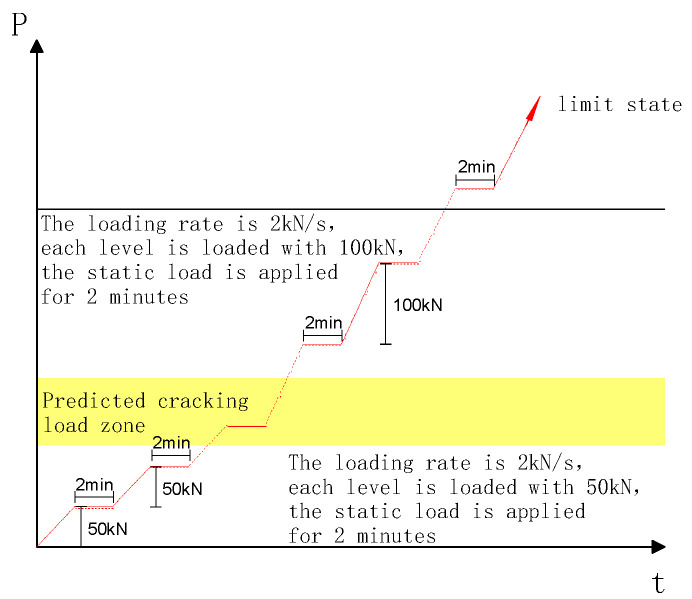
Experimental loading system curve.

**Figure 5 materials-16-06023-f005:**
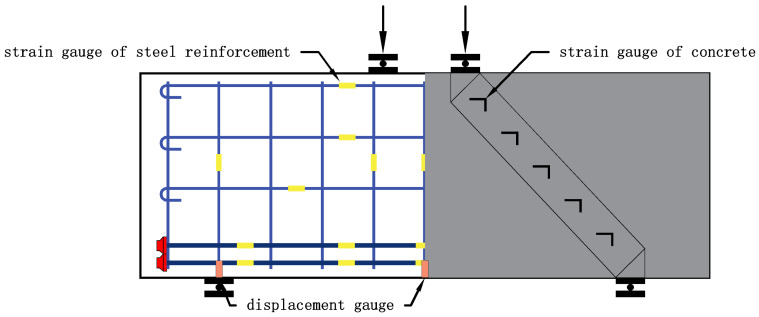
Arrangement of displacement meters and strain gauges.

**Figure 6 materials-16-06023-f006:**
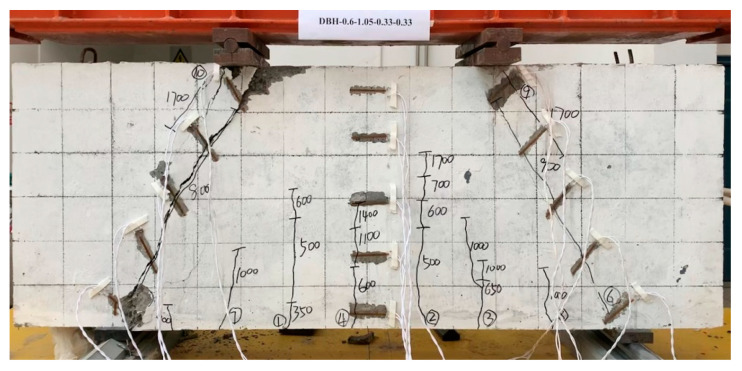
DBH-2 failure mode.

**Figure 7 materials-16-06023-f007:**
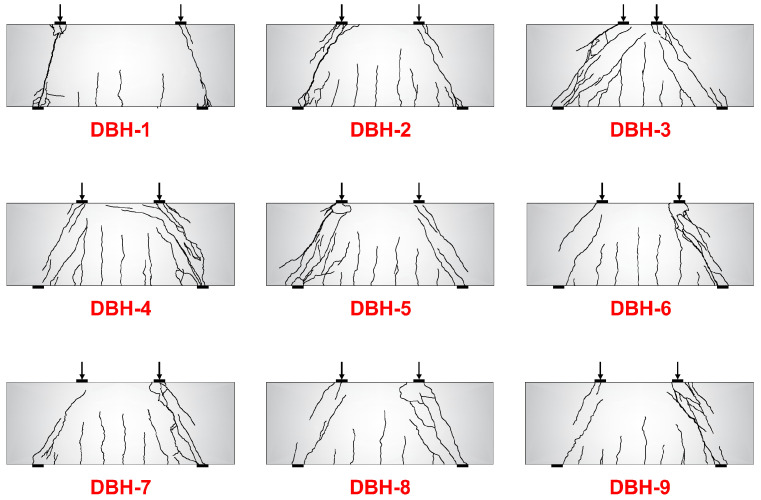
Crack failure mode of specimens.

**Figure 8 materials-16-06023-f008:**
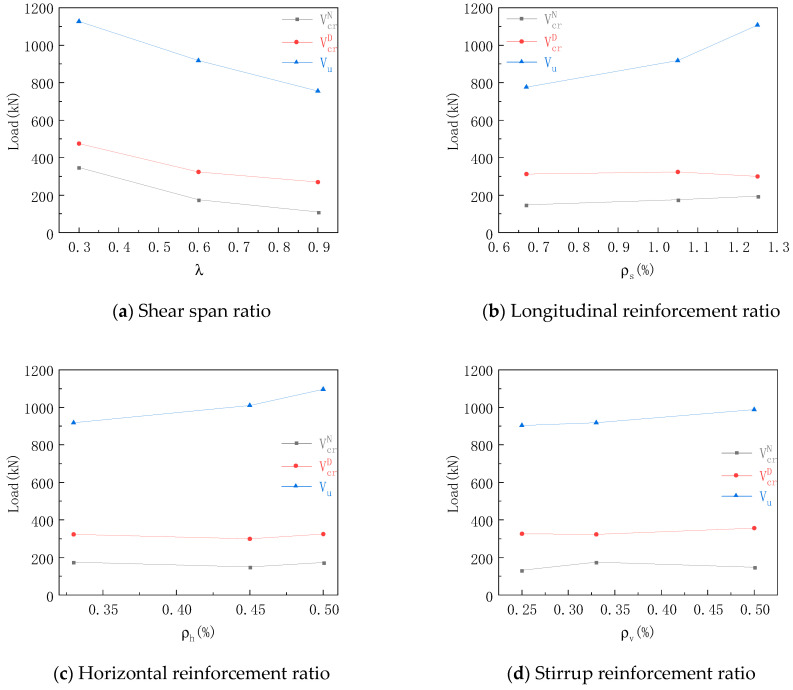
Characteristic load curves.

**Figure 9 materials-16-06023-f009:**
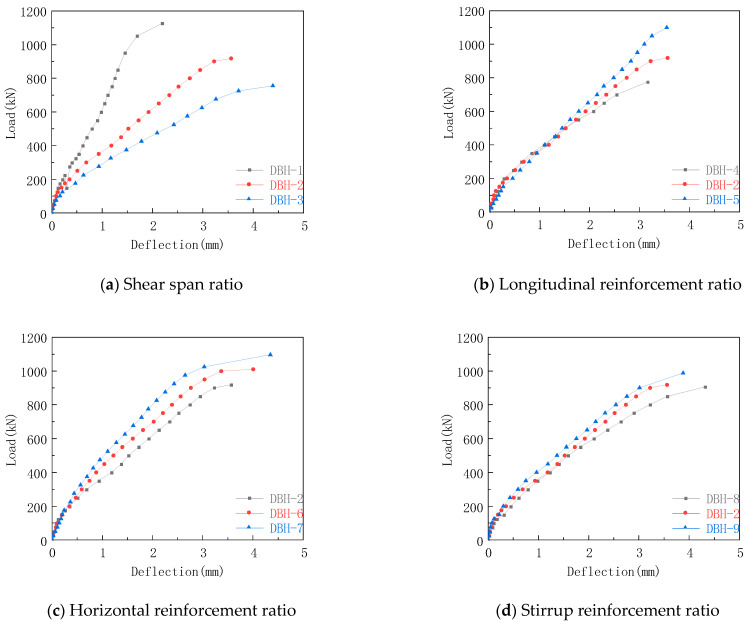
Mid-span deflection curves of each group.

**Figure 10 materials-16-06023-f010:**
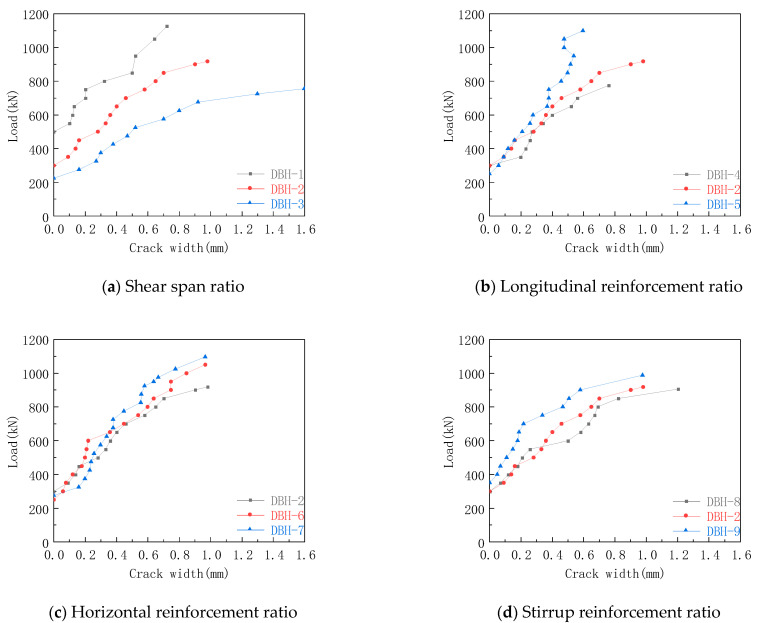
Crack width curve of oblique section.

**Figure 11 materials-16-06023-f011:**
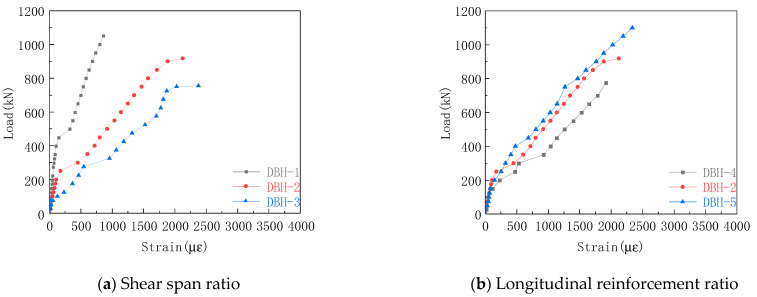
Average strain curve of longitudinal reinforcement.

**Figure 12 materials-16-06023-f012:**
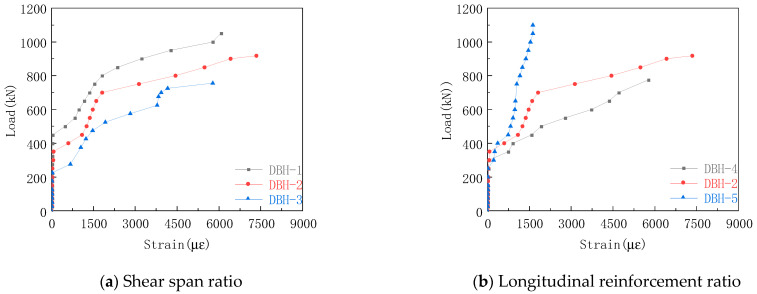
Average strain curve of horizontal reinforcement.

**Figure 13 materials-16-06023-f013:**
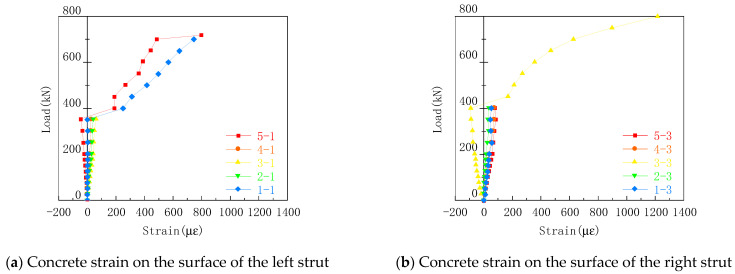
DBH-2 concrete strain curve.

**Figure 14 materials-16-06023-f014:**
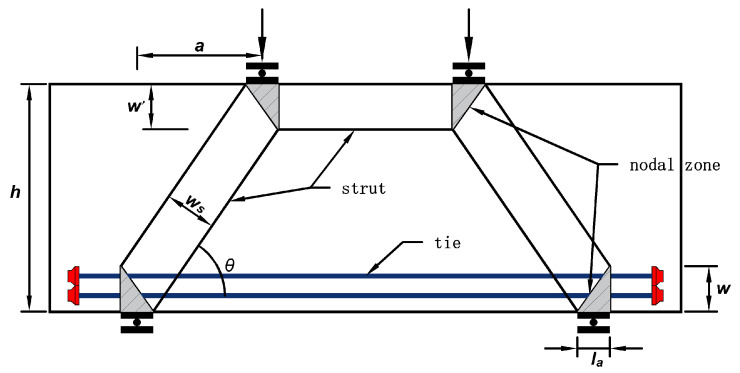
STM for deep beam.

**Figure 15 materials-16-06023-f015:**
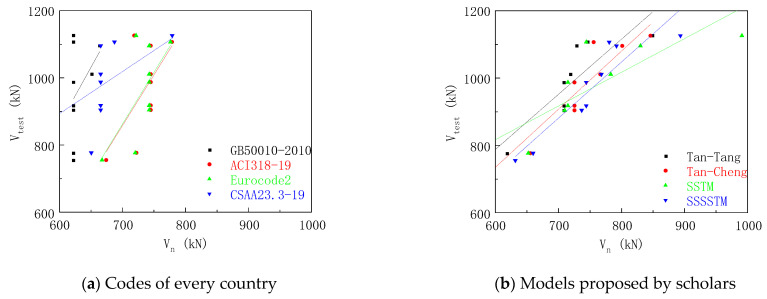
Comparison of results.

**Table 1 materials-16-06023-t001:** Main design parameters of specimens.

Specimen Number	Parameters of the Specimen	λ	Longitudinal Reinforcement	$\rho_{s}$/%	Horizontal Reinforcement	$\rho_{h}$/%	Stirrup Reinforcement	$\rho_{v}$/%	Anchor Plate Diameter/mm
DBH-1	DBH-0.3-1.05-0.33-0.33	0.3	4D^T^20	1.05	C8@150	0.33	C8@150	0.33	48
DBH-2	DBH-0.6-1.05-0.33-0.33	0.6	4D^T^20	1.05	C8@150	0.33	C8@150	0.33	48
DBH-3	DBH-0.9-1.05-0.33-0.33	0.9	4D^T^20	1.05	C8@150	0.33	C8@150	0.33	48
DBH-4	DBH-0.6-0.67-0.33-0.33	0.6	4D^T^16	0.67	C8@150	0.33	C8@150	0.33	42
DBH-5	DBH-0.6-1.25-0.33-0.33	0.6	4D^T^22	1.25	C8@150	0.33	C8@150	0.33	52
DBH-6	DBH-0.6-1.05-0.45-0.33	0.6	4D^T^20	1.05	C8@112.5	0.45	C8@150	0.33	48
DBH-7	DBH-0.6-1.05-0.50-0.33	0.6	4D^T^20	1.05	C8@100	0.50	C8@150	0.33	48
DBH-8	DBH-0.6-1.05-0.33-0.25	0.6	4D^T^20	1.05	C8@150	0.33	C8@200	0.25	48
DBH-9	DBH-0.6-1.05-0.33-0.50	0.6	4D^T^20	1.05	C8@150	0.33	C8@100	0.50	48

λ is the shear span ratio; $\rho_{s}$ is the longitudinal reinforcement ratio; $\rho_{h}$ is the horizontal reinforcement ratio; $\rho_{v}$ is the stirrup reinforcement ratio; DT is 600 MPa high-strength reinforcement.

**Table 2 materials-16-06023-t002:** Concrete material performance test results.

Grade of Concrete	$f_{cu}$/MPa	$f_{c}$/MPa	$f_{t}$/MPa	$E_{c}$/GPa
C60	73.16	64.47	3.64	37.1

$f_{cu}$ is cubic compressive strength of concrete; $f_{c}$ is axial compression strength of concrete; $f_{t}$ is tensile strength of concrete; $E_{c}$ is elastic modulus of concrete.

**Table 3 materials-16-06023-t003:** Steel material performance test results.

Category	Reinforcement	d/mm	$f_{y}$/MPa	$f_{u}$/MPa	$E_{c}$/GPa
Longitudinal reinforcement	HRB600	16	670	865	198.5
Longitudinal reinforcement	HRB600	20	653.7	823.3	196.6
Longitudinal reinforcement	HRB600	22	630	800	195.8
Web reinforcement	HRB400E	8	456.8	647.7	205.3

d is reinforcement diameter; $f_{y}$ is specified yield strength for reinforcement; $f_{u}$ is ultimate strength for reinforcement; $E_{c}$ is modulus of elasticity of reinforcement.

**Table 4 materials-16-06023-t004:** The measured value of characteristic loads.

Specimen Number	$V_{cr}^{N}$ /kN	$V_{cr}^{D}$ /kN	$V_{u}$ /kN	$V_{cr}^{N}$ $/V_{u}$	$V_{cr}^{D}$ $/V_{u}$	δ /mm	Diagonal Crack Width/mm	Mode of Failure	Manifestation	Anchor Destroyed
DBH-1	347	475	1126	30.82%	42.18%	2.19	0.72	Diagonal compression failure	Strut rod crushing	No
DBH-2	175	323	918	19.06%	35.19%	3.56	0.98	Diagonal compression failure	Strut rod crushing	No
DBH-3	110	270	755	14.57%	35.76%	4.39	1.60	Diagonal compression failure	Strut rod crushing	No
DBH-4	148	313	776.5	19.06%	40.31%	3.16	0.76	Diagonal compression failure	Strut rod crushing	No
DBH-5	194.5	300	1107	17.57%	27.10%	3.55	0.60	Diagonal compression failure	Strut rod crushing	No
DBH-6	150	300	1011	14.84%	29.67%	4.01	0.97	Diagonal compression failure	Strut rod crushing	No
DBH-7	174	325	1096	15.88%	29.65%	4.35	0.97	Diagonal compression failure	Strut rod crushing	No
DBH-8	132	327	904	14.60%	36.17%	4.31	1.22	Diagonal compression failure	Strut rod crushing	No
DBH-9	149	357	987.5	15.09%	36.15%	3.88	0.98	Diagonal compression failure	Strut rod crushing	No

**Table 5 materials-16-06023-t005:** Codes of every country.

Source of Formula	Computing Formula	Parameter
GB50010-2010 [19]	$V\leq \frac{1.75}{\lambda +1}f_{t}bh_{0}+\frac{\left(\frac{l_{0}}{h}-2\right)}{3}f_{yv}\frac{A_{sv}}{S_{h}}h_{0}$ $+\frac{\left(5-l_{0}/h\right)}{6}f_{yh}\frac{A_{sh}}{S_{v}}h_{0}$	λ is the calculated shear span ratio, $\mathrm{when}\;l_{0}/h$ ≤ 2, λ = 0.25; $f_{t}$ is the design value of axial tensile strength of concrete; b is the width of deep-beam section; $h_{0}$ is the effective section height of deep beam; $l_{0}/h$$\;\mathrm{is}\;\mathrm{the}\;\mathrm{span}-\mathrm{depth}\;\mathrm{ratio}\;\mathrm{of}\;\mathrm{deep}\;\mathrm{beam},\;\mathrm{when}\;l_{0}/h$ ≤ 2, $l_{0}/h$ = 2; $f_{yv}$$\;\mathrm{and}\;f_{yh}$ are the design values of tensile strength of the stirrups and horizontal reinforcement; $A_{sv}$$\;\mathrm{and}\;A_{sh}$ is stirrups and horizontal reinforcement cross-sectional area; $S_{h}$$\;\mathrm{and}\;S_{v}$ are the spacing between stirrups and horizontal bars
ACI318-19 [27]	$V_{n}=0.85\beta_{c}\beta_{s}f_{c}^{,}A_{cs}\sin\theta$ $A_{cs}=b_{w}w_{s}$ $w_{s}=w_{t}\cos\theta +l_{b}\sin\theta$ $\tan\theta =\frac{h-(w_{t}+w_{t}^{,})/2}{a}\geq 0.488$	$V_{n}$ is the shear capacity of deep beam; $f_{c}^{,}$ is the compressive strength of cylindrical concrete; θ is the minimum angle between the concrete strut and the steel tie rod connected to it, and the specification should not be less than 25°; $\beta_{s}$ is the strength reduction coefficient of concrete; $\beta_{c}$ is the correction coefficient; $A_{cs}$ is the cross-sectional area of the strut bar; $b_{w}$ is the width of the specimen section; $w_{s}$ is the width of concrete diagonal strut bar; $w_{t}$$\;\mathrm{and}\;w_{t}^{,}$ are the heights of different strut joints; $w_{t}^{,}=w_{t};\;l_{b}$ is the width of the support plate
EC [28]	$tan\theta =\frac{h-(w_{t}+w_{t}^{\prime \prime })/2}{a}$ $w_{t}^{,}=1.176w_{t}$	The parameter is the same as the American code
CSAA23.3-19 [29]	$f_{cu}=\frac{f_{c}^{,}}{0.8+170\varepsilon_{1}}\leq 0.85f_{c}^{,}$ $\varepsilon_{1}=\varepsilon_{s}+(\varepsilon_{s}+0.002)\cot^{2}\theta_{s}$	$\varepsilon_{s}$ is the tensile strain of the tie bar

**Table 6 materials-16-06023-t006:** Tan–Tang model and Tan–Cheng model.

Source of Formula	Computing Formula	Parameter
Tan–Tang [30]	$\frac{1}{V}=\frac{1}{V_{ds}}+\frac{1}{V_{dc}}$ $V_{dc}=f_{c}^{,}A_{str}\sin\theta_{s}$ $V_{ds}=f_{ct}A_{ct}\frac{1}{2\cos\theta_{s}}+f_{yw}A_{w}\frac{\sin\left(\theta_{s}+\theta_{w}\right)}{2\cos\theta_{s}}$ $+f_{y}A_{s}\tan\theta_{s}$	$f_{c}^{,}$ is the compressive strength of cylindrical concrete; $A_{str}$ is the cross-sectional area of the strut bar; $\theta_{s}$ is the angle of strut bar; $f_{ct}$, $f_{yw},$ and $f_{y}$ are tensile strength of concrete, yield strength of web reinforcement, and yield strength of longitudinal reinforcement.; $A_{ct}$, $A_{w},$ and $A_{s}$ are the cross-sectional area of concrete struts, web reinforcements, and bottom longitudinal reinforcements.; $\theta_{w}$ is the angle between the horizontal reinforcement and the horizontal direction
Tan–Cheng [31]	$V=\frac{1}{\frac{\sin2\theta_{s}}{f_{t}A_{c}}+\frac{1}{f_{c}^{,}A_{str}\sin\theta_{s}}}$ $f_{t}=\frac{2A_{s}f_{y}\sin\theta_{s}}{A_{c}/\sin\theta_{s}}+\sum \frac{2A_{w}f_{yw}\sin\left(\theta_{s}+\theta_{w}\right)}{A_{c}/\sin\theta_{s}}\;\cdot \frac{d_{w}}{h_{0}}+0.5\sqrt{f_{c}^{,}}$ $\xi =0.8+\frac{0.4}{\sqrt{1+(l-s)/50}}$ $\zeta =0.5+\sqrt{\frac{kd_{s}}{s_{w}}}\leq 1.2$	$f_{t}$ is the composite tensile strength; ξ is the correction coefficient considering the geometric size of the strut bar; ζ is the correction coefficient considering the boundary condition of the strut bar.

**Table 7 materials-16-06023-t007:** Calculation results.

Specimen Number	Text Value Vtest/kN	Calculating Value Vn/kN							
		GB50010-2010	ACI318-19	EC2	CSAA23.3-19	Tan–Tang	Tan–Cheng	SSTM	SSSTM
DBH-1	1126	622.41	718.71	721.42	779.22	848.44	845.92	991	893.27
DBH-2	918	622.41	744.74	742.62	665.56	708.69	725.51	715	743.97
DBH-3	755	622.41	674.29	667.47	466.17	582.79	596.19	586	631.64
DBH-4	776.5	622.41	722.25	720.28	650.55	618.79	655.32	652	660.10
DBH-5	1107	622.41	778.72	776.33	687.34	746.22	755.51	744	780.66
DBH-6	1011	651.57	744.74	742.61	665.56	718.87	766.62	783	768.05
DBH-7	1096	663.72	744.74	742.61	665.56	728.61	801.22	830	792.13
DBH-8	904	622.41	744.74	742.61	665.56	708.69	725.51	710	736.97
DBH-9	987.5	622.41	744.74	742.61	665.56	708.69	725.51	715	743.97

**Table 8 materials-16-06023-t008:** Comparison of calculation and experimental results.

Specimen Number	Text Value Vtest/kN	Vtest/Vn							
		GB50010-2010	ACI318-19	EC2	CSAA23.3-19	Tan–Tang	Tan–Cheng	SSTM	SSSTM
DBH-1	1126	1.809	1.566	1.561	1.445	1.327	1.331	1.136	1.261
DBH-2	918	1.474	1.232	1.236	1.379	1.295	1.265	1.283	1.233
DBH-3	755	1.213	1.119	1.131	1.619	1.295	1.266	1.288	1.195
DBH-4	776.5	1.247	1.075	1.078	1.193	1.255	1.185	1.191	1.041
DBH-5	1107	1.778	1.421	1.425	1.611	1.483	1.465	1.488	1.489
DBH-6	1011	1.551	1.357	1.361	1.519	1.406	1.319	1.291	1.316
DBH-7	1096	1.651	1.471	1.475	1.646	1.504	1.368	1.320	1.383
DBH-8	904	1.452	1.213	1.217	1.358	1.276	1.246	1.273	1.226
DBH-9	987.5	1.588	1.325	1.329	1.483	1.393	1.361	1.381	1.327
Mean		1.529	1.309	1.313	1.473	1.359	1.312	1.295	1.282
Variance		0.043	0.027	0.026	0.022	0.007	0.006	0.010	0.007

## Data Availability

Data are contained within the article.

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
