# Peer review of "Experimental Study of Shear Performance of High-Strength Concrete Deep Beams with Longitudinal Reinforcement with Anchor Plate"

_materials, 2023, doi:10.3390/ma16176023_

Round 1

Reviewer 1 Report

The submitted paper presents an interesting recent experimental campaign on concrete deep beams. The novelty is rather limited (or not enough evidenced), but new experimental outcomes are always welcome to enrich existing data-bases.

The results are clearly reported, so the paper is clearly worth being published, but we have some comments.

- Language should be revised. Even the title is unclear: what is a “longitudinal reinforcement of anchored plate”? I guess that it is a reinforcement “with” anchor plates (see how it is correctly said in the literature review…). Please also correct in the entire text.

- The choice of varied parameters is fine, but 9 tests for 4 parameters means that not all combinations are considered. Please motivate the choice of the investigated combinations.

- In general, the fact that the reinforcements are anchored with plates is not enough investigated: what is the difference with conventional anchoring? And how are the anchor plates designed?

- When positioning the paper in the literature (page 3 – lines 110-112), it is said that “little research does exist on high-strength deep beams”. We understand that some research exists. Give a comparison with your own results.

- The specimens are rather small. Please comment on the possible scale effect.

- C60 is not very high-strength. Please compare with the concrete classes used in previous test from other authors.

- In figure 7 and following, specify what is the “load” given on the vertical axis. Total vertical load or shear (or anything else)?

- In figure 8, what happens at the end of the curve? Is it a full “destroying brittle collapse” (if yes, show some pictures)? Or is it decided to stop the test because of too high damage level?

- Figure 10 and 11: clarify the difference between a “longitudinal” and a “horizontal” reinforcement.

- When evaluating the resistance with theoretical models, why considering only force-based or STM, and not also kinematic models, like e.g. proposed by Mihaylov et al ?

See above: main concern is the wording "of anchored plate".

A few minor translation issues that could be easily checked with a careful proof-reading by an English-educated researcher.

Reviewer 2 Report

1.     The abstract provides comprehensive information about the tested samples and the results obtained. However, from a scientific point of view, it would be logical to briefly outline the research problem. So it will be easier for the reader to understand the goals and objectives of the authors, their contribution to science and to correctly evaluate the results obtained.

2.     The discussion of the obtained results is not presented in the form in which it is required in journals of this level. A detailed comparative analysis of the obtained results with the results obtained earlier by other authors should be carried out. The authors must give a detailed comparison, and only then the reader will understand the contribution of the authors to science and scientific novelty, and the new knowledge that has been obtained or the existing ideas that have been developed.

3.     In general, the article is interesting and scientifically developed, but the analytical component should be strengthened. The article is quite promising, but needs to be improved. After corrections, the article can be published.

Reviewer 3 Report

The article deals with the Experimental Study on the Shear Performance of High Strength Concrete Deep Beams with Longitudinal Reinforcement of Anchored Plate. The reviewer appreciates the effort of the authors in carrying out this experimental investigation on deep beams. The manuscript is well-written and presented. The whole design of the research is logical and leads to conclusions.

The reviewer has a few comments on the manuscript to improve the quality and interest of the readers.

1. In Fig.1, the diameter and spacing of reinforcement can be given for better understanding

2. The abbreviation for STM, SSTM & SSSTM can be mentioned initially.

3. The real image of the reinforcement with anchor plates can be shown in the manuscript to gain the reader's interest. 

The language is good and clear. 
